# Mixed Weyl semimetals and low-dissipation magnetization control in insulators by spin–orbit torques

Jan-Philipp Hanke [1], Frank Freimuth [1], Chengwang Niu[1], Stefan Blügel [1] & Yuriy Mokrousov [1]

Reliable and energy-efficient magnetization switching by electrically induced spin–orbit torques is of crucial technological relevance for spintronic devices implementing memory and logic functionality. Here we predict that the strength of spin–orbit torques and the Dzyaloshinskii-Moriya interaction in topologically nontrivial magnetic insulators can exceed by far that of conventional metals. In analogy to the quantum anomalous Hall effect, we explain this extraordinary response in the absence of longitudinal currents as hallmark of monopoles in the electronic structure of systems that are interpreted most naturally within the framework of mixed Weyl semimetals. We thereby launch the effect of spin–orbit torque into the field of topology and reveal its crucial role in mediating the topological phase transitions arising from the complex interplay between magnetization direction and momentum-space topology. The presented concepts may be exploited to understand and utilize magnetoelectric coupling phenomena in insulating ferromagnets and antiferromagnets.

[1] Peter Grünberg Institut and Institute for Advanced Simulation, Forschungszentrum Jülich and JARA, 52425 Jülich, Germany. Correspondence and requests for materials should be addressed to J.-P.H. (email: j.hanke@fz-juelich.de)

Progress in control and manipulation of the magnetization in magnetic materials is pivotal for the innovative design of future nonvolatile, high-speed, low-power, and scalable spintronic devices. The effect of spin–orbit torque (SOT) provides an efficient means of magnetization control by electrical currents in systems that combine broken spatial inversion symmetry and spin–orbit interaction[1–5]. These current-induced torques are believed to play a key role in the practical implementation of various spintronics concepts, since they were demonstrated to mediate the switching of single ferromagnetic layers[6,7] and anti-ferromagnets[8] via the exchange of spin angular momentum between the crystal lattice and the (staggered) collinear magne-tization. Among the two different contributions to SOTs, the so-called antidamping torques are of utter importance owing to the robustness of their properties with respect to details of disorder[5].

Only recently, the research on electrically controlled magneti-zation switching started to reach out to topological condensed matter—for example, very efficient magnetization switching has been achieved lately in metallic systems incorporating topological insulators[9]. Although in latter cases a strong torque can be gen-erated, the resulting electric-field response does not rely on the global topological properties of these trivial systems. The dis-covery of a quantized version of the anomalous Hall effect in magnetic insulators with nontrivial topology in momentum space[10–12] led to a revolution in forging new spintronic device concepts that utilize topology. On the other hand, moving the field of magnetization control by SOTs into the realm of topo-logical spintronics would open bright avenues in exploiting universal arguments of topology for designing magnetoelectric coupling phenomena in magnetic insulators.

With this work, we firmly put the phenomenon of SOT on the topological ground. Employing theoretical techniques we investigate the origin and size of antidamping SOTs and Dzyaloshinskii-Moriya interaction (DMI) in prototypes of topologically nontrivial magnetic insulators, demonstrate that complex topological properties have a direct strong impact on the emergence and magnitude of SOT and DMI in various classes of magnetic insulators, and formulate intriguing perspectives for the electric-field control of magnetization in the absence of longitudinal charge currents.

## Results

**Mixed Weyl semimetals and SOT.** In a clean sample, the anti-damping SOT **T** acting on the magnetization in linear response to the electric field **E** is mediated by the so-called torkance tensor $\tau$, i.e., $\mathbf{T} = \tau\mathbf{E}$[13] (see Fig. 1a, b). The Berry phase nature of the antidamping SOT manifests in the fact that the tensor elements $\tau_{ij}$ are proportional to the mixed Berry curvature $\Omega_{ij}^{\hat{\mathbf{m}}\mathbf{k}} = \hat{\mathbf{e}}_i \cdot 2\,\mathrm{Im}\sum_n^{\mathrm{occ}}\langle\partial_{\hat{\mathbf{m}}}u_{\mathbf{k}n}|\partial_{k_j}u_{\mathbf{k}n}\rangle$ of all occupied states[13–15] which incorporates derivatives of lattice-periodic wave functions $u_{\mathbf{k}n}$ with respect to both crystal momentum **k** and magnetization direction $\hat{\mathbf{m}}$. Here, $\hat{\mathbf{e}}_i$ denotes the $i$th Cartesian unit vector. Inti-mately related to the antidamping SOT is the DMI[16,17] crucial for the emergence of chiral domain walls and chiral skyrmions[18–21] which can be quantified by the so-called spiralization tensor $D$

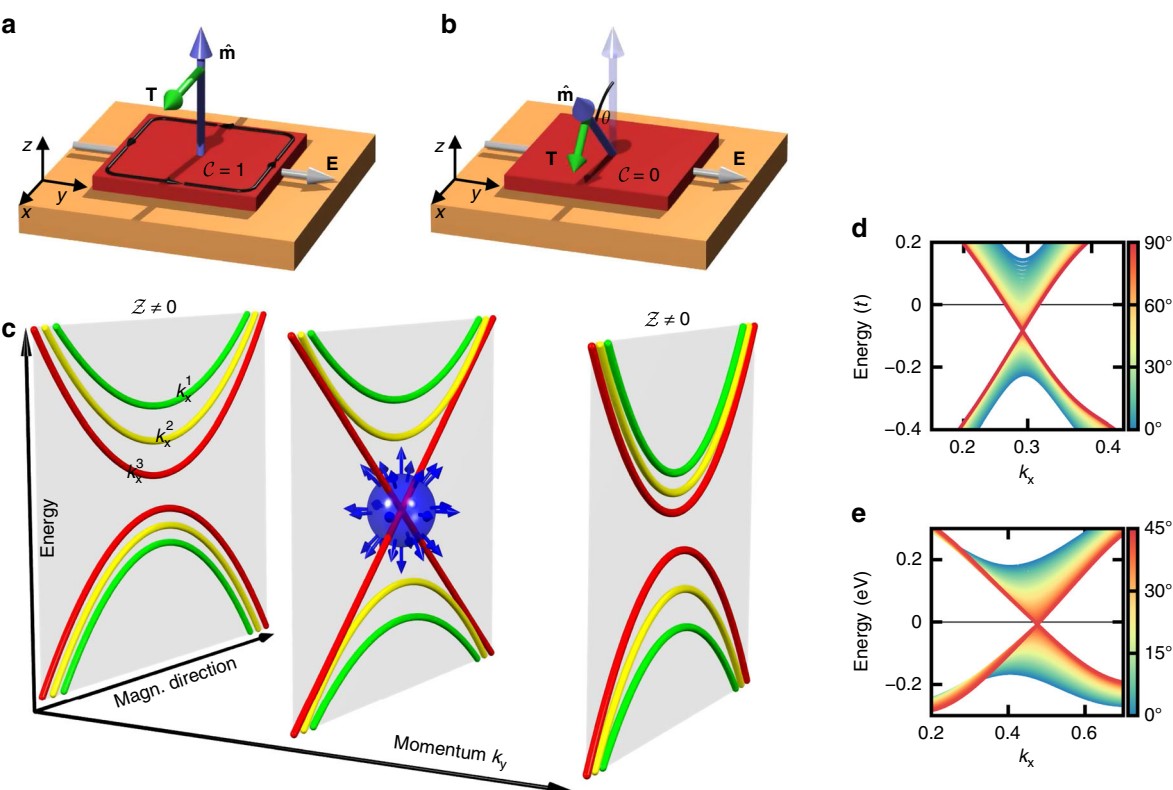

**Fig. 1** Emergence of mixed Weyl points. **a** The magnetization $\hat{\mathbf{m}}$ of a topologically nontrivial insulator is subject to the antidamping torque **T** if an electric field **E** is applied. **b** The resulting reorientation of the magnetization by $\theta$ can trigger a topological phase transition to the trivial insulator. **c** Schematic evolution of two energy bands in the complex phase space of crystal momentum and magnetization direction, where the colors of the bands indicate different $k_x$. If $k_y$ is tuned, the electronic structure displays a monopole, which is correlated with a change in the mixed Chern number $\mathcal{Z}$. Such crossing points are observed in **d** the model of magnetically doped graphene with hopping $t$, and **e** the functionalized bismuth film, where colors indicate the magnetization direction $\hat{\mathbf{m}} = (\sin\theta, 0, \cos\theta)$. The shown monopoles arise at $\theta = 90°$ and $\mathbf{k} = \left(0.29\frac{2\pi}{a_x}, 0.41\frac{2\pi}{a_y}\right)$ for (**d**), and $\theta = 43°$ and $\mathbf{k} = (0.48, 0.19)$ in internal units for (**e**)

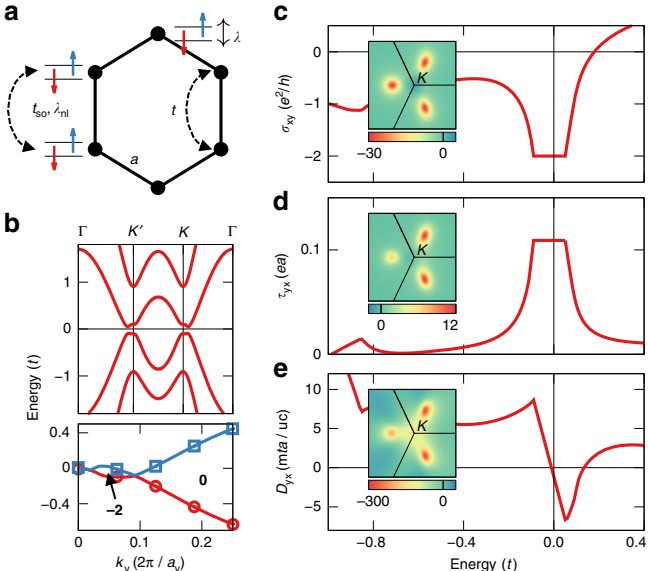

**Fig. 2** Model of magnetically doped graphene. **a** Sketch of the tight-binding model. **b** Band structure with out-of-plane magnetization and $t_{so} = 0.3\,t$, $\lambda = 0.1\,t$, and $\lambda_{nl} = 0.4\,t$. In addition, we show the valence band maximum (red circles) and conduction band minimum (blue squares) for given values of $k_y$, where the depicted gap closing occurs for $\theta = 270°$ and $\mathbf{k} = \left(0.71\frac{2\pi}{a_x}, 0.09\frac{2\pi}{a_y}\right)$. The bold integers denote the mixed Chern number $\mathcal{Z}$ in the insulating regions, and $a_y = 3a/2$. **c–e** Energy dependence of the anomalous Hall conductivity $\sigma_{xy} = \mathcal{C}e^2/h = e^2/(2\pi h)\int\Omega_{xy}^{\mathbf{kk}}dk_xdk_y$, the torkance $\tau_{yx}$, and the spiralization $D_{yx}$, respectively, for an out-of-plane magnetization. Insets show the corresponding momentum-space distributions summed over all occupied states in the vicinity of the $K$-point

reflecting the change of the free energy $F$ due to chiral perturbations $\partial_j \hat{\mathbf{m}}$ according to $F = \sum_{ij} D_{ij}\hat{\mathbf{e}}_i \cdot \left(\hat{\mathbf{m}} \times \partial_j\hat{\mathbf{m}}\right)$[13].

Optimizing the efficiency of magnetization switching in spintronic devices by current-induced SOTs relies crucially on the knowledge of the microscopic origin of most prominent contributions to the electric-field response. To promote the understanding, it is rewarding to draw an analogy between the antidamping SOT as given by $\Omega_{ij}^{\mathbf{mk}}$ and the intrinsic anomalous Hall effect as determined by the Berry curvature $\Omega_{ij}^{\mathbf{kk}} = 2\text{Im}\sum_n^{\text{occ}}\langle\partial_{k_i}u_{\mathbf{k}n}|\partial_{k_j}u_{\mathbf{k}n}\rangle$[22]. Both $\Omega^{\mathbf{kk}}$ and $\Omega^{\mathbf{mk}}$ are components of a general curvature tensor $\Omega$ in the composite $(\mathbf{k}, \hat{\mathbf{m}})$ phase space combining crystal momentum and magnetization direction[23,24]. Band crossings, also referred to as magnetic monopoles in $\mathbf{k}$-space, are known[25] to act as important sources or sinks of $\Omega^{\mathbf{kk}}$. When transferring this concept to current-induced torques, crossing points in the composite phase space can be anticipated to give rise to a large mixed Berry curvature $\Omega^{\mathbf{mk}}$, which in turn yields the dominant microscopic contribution to torkance and spiralization. Thus, materials providing such monopoles close to the Fermi energy can be expected to exhibit notably strong SOTs and DMI.

In the field of topological condensed matter[26,27] the recent advances in the realization of quantum anomalous Hall or Chern insulators have been striking[11,12]. These magnetic materials are characterized by a quantized value of the anomalous Hall conductivity and an integer nonzero value of the Chern number in $\mathbf{k}$-space, $\mathcal{C} = 1/(2\pi)\int\Omega_{xy}^{\mathbf{kk}}dk_xdk_y$. On the other hand, topological semimetals have recently attracted great attention due to their exceptional properties stemming from monopoles in momentum space. Among these latter systems, magnetic Weyl semimetals host gapless low-energy excitations with linear dispersion in the vicinity of nondegenerate band crossings at

generic $\mathbf{k}$-points[28–31], which are sources of $\Omega^{\mathbf{kk}}$. Also referred to as Weyl fermions, these quasiparticles are conventionally described by the Hamiltonian $H_w = \sum_i v_i k_i \sigma_i$, where $\boldsymbol{\sigma} = (\sigma_x, \sigma_y, \sigma_z)$ is the vector of Pauli matrices. Besides the on-going intensive efforts in discovering new type-I and type-II Weyl semimetals[29,30,32], scrutinizing the stability and symmetry protection of the Weyl points and uncovering exotic transport properties of the Weyl phase are hot topics of ever-growing research interest[33].

Here, we introduce the concept of a mixed Weyl semimetal by formally replacing one of the momentum variables with the magnetization direction (specified by an angle $\theta$) in the usual Weyl Hamiltonian. This results in the low-energy description of the system in the combined phase space of $\mathbf{k} = (k_x, k_y)$ and $\theta$ by $H_{mw} = v_x k_x \sigma_x + v_y k_y \sigma_y + v_\theta \theta \sigma_z$, where $\theta$ is the angle that the magnetization $\hat{\mathbf{m}} = (\sin\theta, 0, \cos\theta)$ makes with the $z$-axis. By introducing the concept of a mixed Weyl semimetal, we endeavour to generalize the notion of a Weyl point to the case of entangled crystal momentum and magnetization direction variables. A distinct property of a mixed Weyl semimetal is that while it is an insulator for a general value of $\theta$, it exhibits band crossings at the Fermi energy for certain distinct values of the magnetization direction, determined by the symmetry of the system. In other words, as illustrated in Fig. 1c, mixed Weyl semimetals as described by the Hamiltonian $H_{mw}$ feature monopoles in the composite phase space of $\mathbf{k}$ and $\theta$. These monopoles serve as the sources of the general curvature tensor $\Omega$. In analogy to conventional Weyl semimetals[28], we can characterize the topology and detect magnetic monopoles by monitoring the flux of the mixed Berry curvature through planes of constant $k_y$ as given by the integer mixed Chern number $\mathcal{Z} = 1/(2\pi)\int\Omega_{yx}^{\mathbf{mk}}d\theta dk_x$ (Fig. 1c). Taking the general viewpoint of magnetization dynamics in topologically nontrivial materials, here we demontrate the existence of mixed Weyl semimetals and focus on the implications of the corresponding monopoles for magnetoelectric properties, leaving the analysis of symmetry requirements which guarantee their emergence for future studies. In the following, we show that a significant electric-field response near monopoles in mixed Weyl semimetals is invaluable in paving the road towards low-dissipation magnetization control by SOTs[34].

**Magnetically doped graphene.** We begin with a tight-binding model of magnetically doped graphene[35]:

$$H = -t\sum_{\langle ij\rangle\alpha}c_{i\alpha}^\dagger c_{j\alpha} + it_{so}\sum_{\langle ij\rangle\alpha\beta}\hat{\mathbf{e}}_z\cdot(\boldsymbol{\sigma}\times\mathbf{d}_{ij})c_{i\alpha}^\dagger c_{j\beta}$$
$$+\lambda\sum_{i\alpha\beta}(\hat{\mathbf{m}}\cdot\boldsymbol{\sigma})c_{i\alpha}^\dagger c_{j\beta} - \lambda_{nl}\sum_{\langle ij\rangle\alpha\beta}(\hat{\mathbf{m}}\cdot\boldsymbol{\sigma})c_{i\alpha}^\dagger c_{j\beta},$$

(1)

which is sketched in Fig. 2a. Here, $c_{i\alpha}^\dagger$ ($c_{i\alpha}$) denotes the creation (annihilation) of an electron with spin $\alpha$ at site $i$, $\langle\ldots\rangle$ restricts the sums to nearest neighbors, and the unit vector $\mathbf{d}_{ij}$ points from $j$ to $i$. Besides the usual hopping with amplitude $t$, the first line in Eq. 1 contains the Rashba spin–orbit coupling of strength $t_{so}$ originating in the surface potential gradient of the substrate. The remaining terms in Eq. 1 are the exchange energy due to the local ($\lambda$) and nonlocal ($\lambda_{nl}$) exchange interaction between spin and magnetization. Depending on $\hat{\mathbf{m}}$, the nonlocal exchange describes a hopping process during which the spin can flip. The Methods section provides further details on the tight-binding model and its numerical solution.

First, by monitoring the evolution of the mixed Chern number $\mathcal{Z}$ we demonstrate that the above model hosts a mixed Weyl semimetal state. Indeed, as shown in Fig. 2b, the topological index $\mathcal{Z}$ changes from $-2$ to $0$ at a certain value of $k_y$, indicating thus

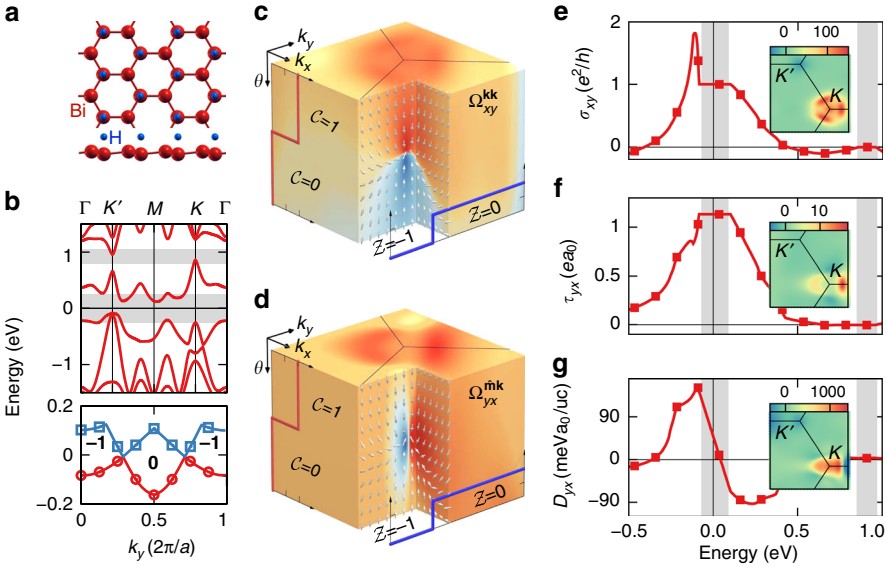

**Fig. 3** Magnetoelectric properties of a mixed Weyl semimetal. **a** Crystal structure of the semi-hydrogenated Bi(111) bilayer. **b** First-principles band structure for an out-of-plane magnetization, where the region of the topologically complex band gap and the trivial one above are highlighted. In addition, the evolution of valence band maximum (red circles) and conduction band minimum (blue squares) with $k_y$ is shown. Bold integers denote the mixed Chern number $\mathcal{Z}$, and $a$ is the in-plane lattice constant. **c**, **d** Monopole-like field of momentum space and mixed Berry curvatures near one of the mixed Weyl points. Arrows indicate the direction of the curvature field ($-\Omega_{yy}^{\bar{m}k}, \Omega_{yx}^{\bar{m}k}, \Omega_{xy}^{kk}$), and a logarithmic color scale is used to display two of its components, where dark red (dark blue) denotes large positive (negative) values. **e**, **g** Energy dependence of $\sigma_{xy}$, $\tau_{yx}$, and $D_{yx}$ for magnetization perpendicular to the film plane. Insets show the microscopic distributions in momentum space near $K$ and $K'$

the presence of a band crossing in composite phase space that carries a topological charge of +2. One of these monopoles appears near the $K$-point off any high-symmetry line if the magnetization is oriented in-plane along the $x$-direction (see Fig. 1d). The emergence of the quantum anomalous Hall effect[35] (Fig. 2c), over a wide range of magnetization directions can be understood as a direct consequence of the magnetic monopoles acting as sources of the curvature $\Omega^{kk}$. Correspondingly, for $\hat{\mathbf{m}}$ out of the plane, the system is a quantum anomalous Hall insulator. Moreover, large values of the mixed curvature $\Omega^{mk}$ in the vicinity of the monopole are visible in the momentum-space distributions of torkance and spiralization in the insets of Fig. 2d, e, respectively. For an out-of-plane magnetization, the primary microscopic contribution to the effects arises from an avoided crossing along $\Gamma K$—a residue of the Weyl point in $(\mathbf{k}, \theta)$-space. Since the expression for the mixed Berry curvature relies only on the derivative of the wavefunction with respect to one of the components of the Bloch vector, the symmetry between $k_x$ and $k_y$ in the distributions of torkance and spiralization is broken naturally (see Methods).

As a consequence of the monopole-driven momentum-space distribution, the energy dependence of the torkance $\tau_{yx}$ (Fig. 2d) displays a decent magnitude of 0.1 $ea$ in the insulating region (with $a$ being the interatomic distance), and stays constant throughout the band gap. In contrast to the Chern numbers $\mathcal{C}$ and $\mathcal{Z}$, the torkance $\tau_{yx}$ is, however, not guaranteed to be quantized to a robust value, i.e., the height of the torkance plateau in Fig. 2d is sensitive to fine details of the electronic structure such as magnetization direction and model parameters. Because of their intimate relation in the Berry phase theory[13,36,37], the plateau in torkance implies a linear behavior of the spiralization $D_{yx}$ within the gap, changing from 8 m$ta$/uc to −6 m$ta$/uc as shown in Fig. 2e, where uc refers to the in-plane unit cell containing two atoms.

To provide a realistic manifestation of the model considerations above, we study from ab initio systems of graphene

decorated by transition-metal adatoms (Fig. 4a). These systems, which exhibit complex spin–orbit mediated hybridization of graphene $p$ states with $d$ states of the transition metal, have by now become one of the prototypical material classes for realization of the quantum anomalous Hall effect[38–42]. Details on the first-principles calculations are provided in the Methods section. In the Chern insulator phase of these materials with magnetization perpendicular to the graphene plane, depending on the transition-metal adatom, both torkance and spiralization can reach colossal magnitudes that originate from mixed Weyl points. In the case of W in 4 × 4-geometry on graphene, for example, the torkance amounts to a very large value of $\tau_{yx} = 2.9$ $ea_0$ (with $a_0$ being Bohr's radius), and the spiralization $D_{yx}$ ranges from −5 meV$a_0$/uc to 60 meV$a_0$/uc (Fig. 4b–e), surpassing thoroughly the values obtained in metallic magnetic hetero-structures[5,13] and non-centrosymmetric bulk magnets[21]. Since the details of the electronic structure can influence the value of the torkance in the gap, upon replacing W with other transition metals, the magnitude of SOT and DMI can be tailored in the gapped regions of corresponding materials according to our calculations.

**Functionalized bismuth film.** Aiming at revealing pronounced magnetoelectric coupling effects in magnetic insulators with larger band gaps as compared to the above examples, we turn to a semi-hydrogenated Bi(111) bilayer (Fig. 3a), which is a prominent example of functionalized insulators realizing nontrivial topological phases[42]. As we show, semi-hydrogenated Bi(111) bilayer is a mixed Weyl semimetal. For an out-of-plane magnetization, the system is a valley-polarized quantum anomalous Hall insulator[43] with a magnetic moment of 1.0 $\mu_{\mathrm{B}}$ per unit cell, and it exhibits a large band gap of 0.18 eV at the Fermi energy as well as a distinct asymmetry between the valleys $K$ and $K'$ (Fig. 3b).

Analyzing the evolution of the mixed Chern number $\mathcal{Z}$ as a function of $k_y$ in Fig. 3b, we detect two magnetic monopoles of

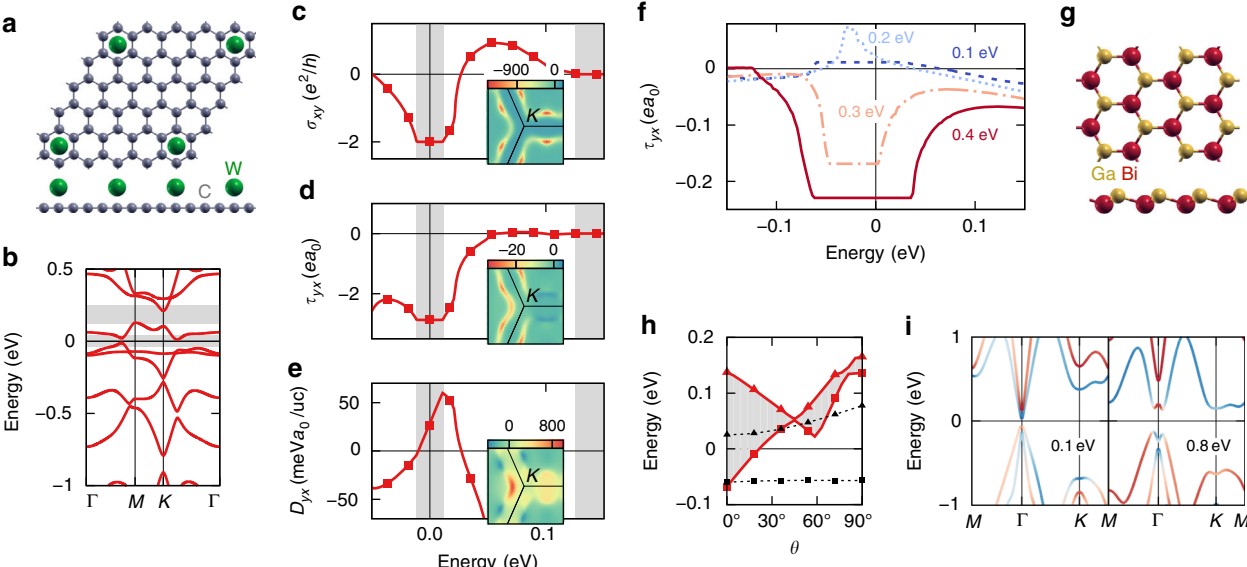

**Fig. 4** Monopole-driven spin–orbit torques in mixed Weyl semimetals. **a** Crystal structure of graphene decorated by W adatoms in 4 × 4 geometry. **b** First-principles band structure for an out-of-plane magnetization. The topologically nontrivial gap around the Fermi level and the trivial gap above are highlighted. **c**–**e** Energy dependence of anomalous Hall conductivity $\sigma_{xy}$, torkance $\tau_{yx}$, and spiralization $D_{yx}$, respectively. Insets show the microscopic distributions in momentum space near the $K$-point. **f** Energy dependence of the torkance $\tau_{yx}$ in a GaBi film upon applying an exchange field $\mathbf{B} = B_0(\sin\theta, 0, \cos\theta)$ perpendicular to the film plane, i.e., $\theta = 0°$. Numbers denote the value of $B_0$. **g** Crystal structure of the system. **h** Evolution of valence band maximum (squares) and conduction band minimum (triangles) with $\theta$ for $B_0 = 0.1$ eV (dashed black) and $B_0 = 0.8$ eV (solid red). **i** Band structures for $\theta = 0°$ and two different values of $B_0$, where colors encode the spin polarization perpendicular to the film

opposite charge that emerge at the transition points between the topologically distinct phases with $\mathcal{Z} = -1$ and $\mathcal{Z} = 0$. Alternatively, these crossing points and the monopole charges in the composite phase space could be identified by monitoring the variation of the momentum-space Chern number $\mathcal{C}$ with magnetization direction. These monopoles occur at generic points near the valley $K$ for $\theta = 43°$ (see Fig. 1e) and in the vicinity of the $K'$-point for $\theta = 137°$, respectively. The presence of such mixed Weyl points in the electronic structure drastically modifies the behavior of the general curvature $\Omega$ in their vicinity, as visible from the three-dimensional representation of $\Omega$ displayed in Fig. 3c, d. Revealing characteristic sign changes when passing through monopoles in composite phase space, the singular behavior of the Berry curvature underlines the role of the mixed Weyl points as sources or sinks of $\Omega$. For an out-of-plane magnetization, the complex nature of the electronic structure in momentum space manifests in the quantization of $\mathcal{C}$ to +1 (Fig. 3e), which is primarily due to the pronounced positive contributions near $K$, where the bands come closest to each other. Calculations of the energy dependence of the torkance and spiralization in the system, shown in Fig. 3f, g, reveal the extraordinary magnitudes of these phenomena of the order of 1.1 $ea_0$ for $\tau_{yx}$ and 50 meV$a_0$/uc for $D_{yx}$, exceeding by far the typical magnitudes of these effects in magnetic metallic materials[5,13,21].

At this point, we would like to comment on the role of the magnetic anisotropy energy for the effects that we study. Our calculations show that the magnetic state of the considered systems is stabilized by anisotropy energies on the order of 1 meV, which is comparable to the values obtained in metallic heterostructures such as Co/Pt. As a consequence of the magnetic anisotropy energy, the magnetization is subject to an additional torque if $\hat{\mathbf{m}}$ is not aligned with the easy axis (or lies outside of the easy plane). This magnetic anisotropy torque is qualitatively very distinct from the electrically controlled antidamping SOT as it is not determined by the Berry curvature, and thus not responsive

to the presence of mixed Weyl points. Performing explicit calculations, we estimate that for the example of the functionalized bismuth film with $\theta = 30°$ the magnitude of the antidamping SOT exceeds the magnetic anisotropy torque if the applied in-plane electric field strength is larger than a relatively small value of 5 mV/Å. The exact relation between the magnetic anisotropy torque and the antidamping SOT can thus be influenced either by tuning the magnitude of the applied in-plane electric field, or by tuning the strength of the magnetic anisotropy barrier via the application of an out-of-plane electric field[39].

**Proof of monopole-driven SOT enhancement.** An important question to ask is whether the colossal magnitude of the SOT in the insulators considered above can be unambiguously identified with the mixed Weyl semimetallic state. In the following, we answer this question by explicitly demonstrating the utter importance of the emergent mixed monopoles for driving pronounced magnetoelectric response. First, by removing the mixed Weyl points from the electronic structure of the model (1) via, e.g., including an intrinsic spin–orbit coupling term, we confirm that the electric-field response is strongly suppressed, which promotes the monopoles as unique origin of large SOT and DMI. Second, to verify this statement from the first-principles calculations, we analyze the electric-field response throughout the topologically trivial gaps above the Fermi level that are highlighted in Figs. 3b and 4b. Since these gaps do not exhibit the mixed Weyl points, we obtain a greatly diminished magnitude of the torkance $\tau_{yx}$ within these energy regions as apparent from Figs. 3f and 4d.

Finally, we clearly demonstrate the key role of these special points by studying an illustrative example: a thin film of GaBi with triangular lattice structure (Fig. 4g). The initial system is a nonmagnetic trivial insulator, on top of which we artificially apply an exchange field $\mathbf{B} = B_0(\sin\theta, 0, \cos\theta)$, with the purpose of

triggering a topological phase transition as a function of the exchange field strength, see Supplementary Note 1. When tuning the exchange field strength $B_0$ we carefully monitor the evolution of the system from a trivial magnetic insulator for $|B_0| \leq 0.2\,\text{eV}$ to a mixed Weyl semimetal as indicated by the emergence of magnetic monopoles in the electronic structure. The latter phase is accompanied by the quantum anomalous Hall effect prominent for a finite range of directions $\theta$, for instance, if **B** is perpendicular to the film plane (Fig. 4h, i). Comparing in Fig. 4f the electric-field response for these two distinct phases, we uniquely identify drastic changes in sign and magnitude of the torkance $\tau_{yx}$ with the transition from the trivial insulator to the mixed Weyl semimetal hosting monopoles near the $\Gamma$-point. This proves the crucial relevance of emergent monopoles in driving magnetoelectric coupling effects in topologically nontrivial magnetic insulators.

## Discussion

Remarkably, the magnetization switching via antidamping torques in mixed Weyl semimetals can be utilized to induce topological phase transitions from a Chern insulator to a trivial magnetic insulator mediated by the complex interplay between magnetization direction and momentum-space topology in these systems as illustrated in Fig. 1a, b. In the case of the functionalized bismuth film, for instance, the material is a trivial magnetic insulator with a band gap of 0.25 eV if the magnetization is oriented parallel to the film plane. Nevertheless, the resulting antidamping torkance in this trivial state is still very large, and the DMI exhibits a strong variation within the gap, see Supplementary Note 2 and Supplementary Figs. 1 and 2. We therefore motivate experimental search and realization of large magnetoelectric response and topological phase transitions in quantum anomalous Hall systems fabricated to date[12,44–46]. Overall, mixed Weyl semimetals that combine exceptional electric-field response with a large band gap (such as, e.g., functionalized bismuth films) lay out extremely promising vistas in room-temperature applications of magnetoelectric coupling phenomena for low-dissipation magnetization control—a subject which is currently under extensive scrutiny (see, e.g., refs. [34,47,48]). In contrast to the antidamping SOT in magnetic metallic bilayers (such as Co/Pt) for which large spin–orbit interaction in the nonmagnetic substrate is necessary for generating large spin Hall effect and large values of SOT[4], the magnitude of the SOT in insulating phases of a mixed Weyl semimetal is driven by the presence of the mixed monopole rather than the spin–orbit strength itself. This opens perspectives in exploiting a strong magnetoelectric response of weak spin–orbit materials.

In the examples that we considered here, the nontrivial topology of mixed Weyl semimetals leads to DMI changes over a wide range of values throughout the bulk band gap, implying that proper electronic structure engineering enables us to tailor both strength and sign of the DMI in a given system, for instance, by doping or applying strain. Such versatility could be particularly valuable for the stabilization of chiral magnetic structures such as skyrmions in insulating ferromagnets. In the latter case, very large values of the antidamping SOT arising in these systems would open exciting perspectives in manipulation and dynamical properties of chiral objects associated with minimal energy consumption by magnetoelectric coupling effects. Generally, we would like to remark that magnetic monopoles in the composite phase space, which we discuss here, do not only govern the electric-field response in insulating magnets but are also relevant in metals, where they appear on the background of metallic bands. Ultimately, in analogy to the (nonquantized) anomalous Hall effect in metals, this makes the analysis of SOT and DMI in metallic systems very complex owing to competing contributions

to these effects from various bands present at the Fermi energy. In addition, the electric-field strength in metals is typically much smaller, limiting thus the reachable magnitude of response phenomena as compared to insulators.

At the end, we reveal the relevance of the physics discussed here for antiferromagnets (AFMs) that satisfy the combined symmetry of time reversal and spatial inversion. SOTs in such AFMs are intimately linked with the physics of Dirac fermions, which are doubly degenerate elementary excitations with linear dispersion[49,50]. In these systems, the reliable switching of the staggered magnetization by means of current-induced torques has been demonstrated very recently[8]. In analogy to the concept of mixed Weyl semimetals presented here, we expect that the notion of mixed Dirac semimetals in a combined phase space of crystal momentum and direction of the staggered magnetization vector will prove fruitful in understanding the microscopic origin of SOTs in insulating AFMs. Following the very same interpretation that we formulated here for ferromagnets, monopoles in the electronic structure of AFMs can be anticipated to constitute prominent sources or sinks of the corresponding general non-Abelian Berry curvature, whose mixed band diagonal components correspond to the sublattice-dependent antidamping SOT, in analogy to the spin Berry curvature for quantum spin Hall insulators and Dirac semimetals[51–53]. Correspondingly, exploiting the principles of electronic structure engineering for topological properties depending on the staggered magnetization could result in an advanced understanding and utilization of pronounced magnetoelectric response in insulating AFMs.

## Methods

**Berry phase expressions for torkance and spiralization**. In order to characterize the antidamping SOTs, we evaluate within linear response the torkance[13]

$$\tau_{ij} = \frac{2e}{N_{\mathbf{k}}}\hat{\mathbf{e}}_i \cdot \sum_{kn}^{\text{occ}}\left[\hat{\mathbf{m}} \times \text{Im}\langle \partial_{\hat{\mathbf{m}}} u_{\mathbf{k}n}|\partial_{k_j} u_{\mathbf{k}n}\rangle\right], \qquad (2)$$

where $N_{\mathbf{k}}$ is the number of **k**-points, and $e > 0$ denotes the elementary positive charge. Similarly, the spiralization[13] is obtained as

$$D_{ij} = \frac{\hat{\mathbf{e}}_i}{N_{\mathbf{k}}V} \cdot \sum_{\mathbf{k}n}^{\text{occ}}\left[\hat{\mathbf{m}} \times \text{Im}\langle \partial_{\hat{\mathbf{m}}} u_{\mathbf{k}n}|h_{\mathbf{k}n}|\partial_{k_j} u_{\mathbf{k}n}\rangle\right], \qquad (3)$$

where $h_{\mathbf{k}n} = H_{\mathbf{k}} + \varepsilon_{\mathbf{k}n} - 2\varepsilon_{\text{F}}$, $H_{\mathbf{k}}$ is the lattice-periodic Hamiltonian with eigenenergies $\varepsilon_{\mathbf{k}n}$, $\varepsilon_{\text{F}}$ is the Fermi level, and $V$ is the unit cell volume.

**Tight-binding calculations**. To arrive at the model Hamiltonian (1), the model in ref. [35] has been generalized to account for arbitrary magnetization directions $\hat{\mathbf{m}}$ and the nonlocal exchange interaction. We obtained a $4 \times 4$ matrix representation of the resulting Hamiltonian on the bipartite lattice of graphene by introducing four orthonormal basis states $|N\alpha\rangle$ that describe electrons with spin $\alpha = \{\uparrow, \downarrow\}$ on the sublattice $N = \{A, B\}$. Using Fourier transformations, we transformed this matrix to a representation $H(\mathbf{k})$ in momentum space, which was subsequently diagonalized at every **k**-point to access the electronic and topological properties. The model parameters $t_{\text{so}} = 0.3\,t$, $\lambda = 0.1\,t$, and $\lambda_{\text{nl}} = 0.4\,t$ were employed in this work. We chose the magnetization direction as $\hat{\mathbf{m}} = (\sin\theta, 0, \cos\theta)$ for a direct comparison between the model and the first-principles calculations.

**First-principles electronic structure calculations**. Using the full-potential linearized augmented plane-wave code FLEUR (see www.flapw.de), we performed self-consistent density functional theory calculations of the electronic structure of (1) graphene decorated with W adatoms in $4 \times 4$ geometry, and (2) a semi-hydrogenated Bi(111) bilayer. The structural parameters of refs. [39,43] were assumed in the respective cases. While we used here the PBE (Perdew–Burke-Ernzerhof) exchange and correlation functional, other choices led to the same mixed Weyl points but at slightly different positions. The effect of spin–orbit coupling was treated within the perturbative second-variation scheme.

Starting from the converged charge density, the Kohn–Sham equations were solved on an equidistant mesh of $8 \times 8$ **k**-points ($6 \times 6$ in case (1)) for 8 different magnetization directions $\hat{\mathbf{m}} = (\sin\theta, 0, \cos\theta)$, where the angle $\theta$ covers the unit circle once. Based on the resulting wave-function information in the composite phase space, we constructed a single set of higher-dimensional Wannier functions[54] (HDWFs) for each of the systems by employing our extension of the wannier90 code[55]. In case (1), we generated 274 HDWFs from 360 bands with the

frozen window up to 4 eV above the Fermi level, and in the case (2), we extracted from 28 bands 14 HDWFs for a frozen window that extends to 2 eV above the Fermi energy.

We used the Wannier interpolation[56,57] that we generalized to treat crystal momentum and magnetization direction on an equal footing[54] in order to evaluate the Berry curvatures $\Omega^{kk}$ and $\Omega^{mk}$. Taking into account the above parametrization of the magnetization direction by $\theta$, we were thereby able to access efficiently the anomalous Hall conductivity $\sigma_{ij}$, the torkance $\tau_{yj}$, and the spiralization $D_{yj}$. Convergence of these quantities was achieved using $1024 \times 1024$ $\mathbf{k}$-points in the Brillouin zone. We obtained the mixed Chern number $\mathcal{Z}(k_y) = 1/(2\pi)\int 2\mathrm{Im}\sum_n^{\mathrm{occ}} \langle \partial_\theta u_{kn} | \partial_{k_x} u_{kn} \rangle d\theta dk_x$ by integrating the mixed Berry curvature on a uniform mesh of 1024 $k_x$-values and 512 angles $\theta$ in [0, $2\pi$].

**Data availability**. The tight-binding code and the data that support the findings of this study are available from the corresponding authors on request.

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

## Acknowledgements

We gratefully acknowledge computing time on the supercomputers JUQUEEN and JURECA at Jülich Supercomputing Center as well as at the JARA-HPC cluster of RWTH Aachen, and funding from the German Research Foundation (Deutsche Forschungsgemeinschaft) under Grant No. MO 1731/5-1 and SPP 1666. We further acknowledge funding from the European Unions Horizon 2020 research and innovation programme under grant agreement number 665095 (FET-Open project MAGicSky). This work has been also supported by the Deutsche Forschungsgemeinschaft (DFG) through the Collaborative Research Center SFB 1238.

## Author contributions

J.-P.H. uncovered the mixed Weyl points as origin of large magnetoelectric coupling effects through model considerations and first-principles calculations. J.-P.H. and Y.M. wrote the manuscript. All authors discussed the results and reviewed the manuscript.

## Additional information

**Competing interests:** The authors declare no competing financial interests.

