## [Peer Review File · Nature Communications]

Reviewers' comments:

Reviewer #1 (Remarks to the Author):

The manuscript presents an interesting and highly original idea on how to maximize spin-orbit torque (SOT), while minimizing dissipation, by exploiting magnetic topological insulators. The SOT has been a major topic in spintronics over the past 10 years, and the present manuscript sheds new light on its origin and control using the proposed systems where the authors connect it to magnetic monopoles in composite k and θ space. The manuscript is carefully written and supported by both model and first-principles calculations on several different realistic systems.

However, before publication some important points, especially for experimentalists, should be explained in greater detail:

1. It is not really clear how the experiment depicted in Fig. 1(a) would be realized in practice to achieve "dissipationless" switching advertised in the title of the manuscript. Do authors have in mind that one takes wire of "mixed Weyl semimetal", connects it to electrodes and injects current into it, or something else? Figure 1(a) and bulk calculations assume infinite material with applied electric field, but in practice device in Fig. 1(a) of finite size will have longitudinal current flowing along the device edges and therefore finite resistance (e.g., $C=1$ means resistance $R=h/2e^2$ in finite size device). For example, even voltage induced precessional switching in conventional CoFeB/MgO junctions with voltage controlled magnetic anisotropy, where thick MgO barrier really prevents flow of longitudinal current, always has nonzero switching energy ~ 10 fJ.
2. "Mixed Weyl semimetal" is not a standard concept or terminology, compared to widely used Weyl semimetals of type I and type II. On the other hand, the authors devote only one inline equation and second row in Fig. 1 to it, which is difficult to follow for either "spintronics" or "topological" community reader.
3. The manuscript suggest to search for possible realizations of predicted effect among quantum anomalous Hall (QAH) systems fabricated to date, and they also advertise "mixed Weyl semimetals lay out extremely promising vistas in room-temperature applications" where room temperature is indeed crucial for applied spintronics. However, all QAH systems fabricated to date operate far below room temperature due to either too small gap driven by spin-orbit coupling or, most importantly, too small Curie temperature for quasi-2D magnets.

4. The topological phase transition, from topologically non-trivial to trivial phase, as a function of magnetization direction is discussed in the manuscript and is illustrated by example in Supplementary Figure 1 assuming that magnetization in the plane leads to trivial phase. However, this is not valid for arbitrary system where in-plane magnetization together with broken symmetries can lead to topologically non-trivial phase, see Phys. Rev. B 94, 085411 (2016).

Reviewer #2 (Remarks to the Author):

Please find attached report

Reviewer #3 (Remarks to the Author):

This manuscript reports a potentially interesting concept for generating large spin-orbit torque and DMI in mixed Weyl semimetals. The SOT and DMI have been calculated within linear response frame by evaluating Berry phase for materials include W-doped Graphene and H-GaBi. It is found that dependent on the magnetic axis when the two materials are in the topology phase a large SOT and DMI will arise in both materials. The concept of generating huge SOT and DMI in insulating mixed Weyl materials seems attractive for spintronics application and the manuscript is therefore can be published in nat.comm.

However there are some concerns if the authors can address.

1. The authors did not discuss the magnetic anisotropy of the proposed materials for example what is the magnetic ground state of the W-doped Graphene and H-GaBi? In the cited paper (PRL 108, 056802), it is shown the W-doped Graphene will show in-plane magnetic anisotropy which means its ground state is trivial and not the mixed Weyl phase. Then experimentally how to make the proposed materials into mixed Weyl semimetals phase? An extra out-of-plane magnetic field or gate voltage will be needed to drive the materials into topological phase? As it is shown in the paper by Zhang et.al(PRL 108, 056802), by using an extra gate voltage may be a better choice. It is better if this point can be addressed in the revised manuscript.
2. Another issue is about the torque produced by magnetic anisotropy. At as we know there is also a torque produced by the magnetic anisotropy when the magnetic axis is rotating away from the easy axis. What's the relation between this magnetic anisotropy torque and the SOT discussed in the present manuscript?
3. The torque has been calculated when the electric field is applied in-plane and in principle there

is also SOT when electric field is vertically applied as in Zhang's paper. Then under which electric field the SOT will be more efficient?

4. A minor comment. It is better that the Cartesian coordinates are shown in Fig. 1a and b, which may describe the directions of electric field, torque, DMI more clearly. And also the torque direction in Fig1b is shown to be in the xz plane, however in the supplementary (Fig.1) the torque components are in xy plane?

Reviewer #2 Comments:

Hanke, *et al* in their manuscript NCOMMS-17-09884 “*MixedWeyl semimetals and dissipationless magnetization control in insulators by spin-orbit torques*” employed both tight-binding models and ab initio electronic structure calculations for the following two-dimensional systems: (1) Magnetically doped graphene; (2) Semi-hydrogenated Bi (111) bilayer; and (3) GaBi thin film of triangular structure under an external magnetic field. In the first two cases, the transition from the magnetic trivial to non-trivial insulator is induced by varying the direction of the magnetization while in the latter the transition from the non-magnetic trivial to a magnetic non-trivial insulator is induced by the external magnetic field. The authors show that the strength of the spin-orbit torque and the Dzyaloshinskii-Moriya interaction in the non-topologically non-trivial magnetic insulator is enhanced by about a factor of 4 to 5 compared to the ferromagnetic/heavy metal bilayers. The authors propose that the underlying mechanism lies on the emergence of **mixed** Weyl semimetal phase in which the magnetic monopoles appear in the composite momentum direction-momentum space topology. The magnetization switching via spin-orbit torque has attracted a lot of attention recently because it has great potential in ultralow-power magnetic random access memory devices. In particular, I find it very interesting that magnitude of the spin orbit torque in insulating phases of a mixed Weyl semimetal is driven by the presence of the mixed monopole rather than the spin-orbit strength itself.

The results are interesting and the proposed underlying mechanism is novel. This work hopefully will stimulate more experimental and theoretical research and has the potential to increase the spin-orbit torque efficiency more than 3-5 times. Consequently, the manuscript does warrant publication in Nature Communications after the authors respond to the following comments and revise the manuscript accordingly.

1. The authors state in the Introduction that “*The Berry phase nature of the anti-damping SOT manifests in the fact that the tensor elements τ_{ij} are proportional to the mixed Berry curvature of all occupied states [13,14], which incorporates derivatives of lattice-periodic wave functions u_{k_n} with respect to both crystal momentum k and magnetization direction \mathbf{m} .*” The authors should also refer to Lee et al, Phys. Rev. **B 91**, 144401 (2015) where the mixed Berry curvature over all occupied states is derived in Eqs. (14) and (15). The interesting point addressed in the above paper is that the authors show that the **Fermi sea contribution** can be converted to a form where the net contribution is evaluated only at the **Fermi surface [Eq. (16)]**. Can the authors comment on this and revise the manuscript accordingly. Namely, one would expect that only the states around the Fermi surface contribute to the current-induced anti-damping torque and not states far away from the Fermi surface.
2. For the model of magnetically doped graphene that authors state “One of these monopoles appears near the K-point off any high symmetry line if the magnetization is oriented **in-plane** along the x-direction”. This is also shown in Fig. 1(d) where the monopole appears at $\theta=90$ and $(k_x, k_y) = (0.29\frac{2\pi}{a_x}, 0.41\frac{2\pi}{a_y})$. Thus, in Fig. 2©, (d) and (f) where the magnetization is **out of plane there are no mixed Weyl nodes**. How, can the authors claim that the large torque of $0.1 ea_0$ in the insulating phase arises from the monopole-driven momentum-space distribution? Indeed the torque shows three-fold symmetry peaks around the K point but the mixed Weyl points occur only if $\theta=90$.

3. The statement in the caption of the bottom of Fig. 2(b) states "... in the (k_x, θ) -space. What are the values of k_x and θ where the gap closes at $k_y = 0.08 \frac{2\pi}{a_y}$ in the figure? The authors should clarify in the main text the statement " (k_x, θ) -space".
4. In order to determine the **accurate** evolution of the energy gap using the ab initio calculations one needs an exchange correlation functional which yields accurate values of the band gap (for example MBJLDA). I assume that the authors employed the PBE functional and hence the emergence of the Weyl points in mixed space is not very accurate. The authors should mention the type of exchange correlation they used and a statement that the emergence of the Weyl points may not be that accurate.
5. It is interesting that the Dzyaloshinskii-Moriya interaction shows a linear variation with energy in all Figs. 1(d), 2(f) and 4(e) and changes sign where the torkance is constant within the specific energy window. What is the underlying origin?
6. Is the Ω_{yy}^{mk} in the caption of Fig. 3 a typo? Why this is not shown in the figure?
7. The statement "...torkance amounts to a **huge** value of $-2.9 ea_0$..." needs to be tone down. A factor of five is not huge.

Response to Reviewers:

We would like to thank the three reviewers for their critical reading as well as for providing valuable remarks that helped us to improve the manuscript. In the following, we present our point-by-point response to all raised issues.

Overview of changes based on reviewers' comments:

- Clarified idea of dissipationless switching
- Clarified nature of mixed Weyl semimetals
- Citation of the suggested Ref. [15]
- Included explicit position of mixed Weyl points in Fig. 2 (b)
- Statement on choice of exchange and correlation functional
- Toned down language at one point
- Included discussion of magnetic anisotropy torque
- Provided coordinate system in Figs. 1 (a) and (b)

Reviewer #1: The manuscript presents an interesting and highly original idea on how to maximize spin-orbit torque (SOT), while minimizing dissipation, by exploiting magnetic topological insulators. The SOT has been a major topic in spintronics over the past 10 years, and the present manuscript sheds new light on its origin and control using the proposed systems where the authors connect it to magnetic monopoles in composite \mathbf{k} and θ space. The manuscript is carefully written and supported by both model and first-principles calculations on several different realistic systems.

However, before publication some important points, especially for experimentalists, should be explained in greater detail:

It is not really clear how the experiment depicted in Fig. 1(a) would be realized in practice to achieve "dissipationless" switching advertised in the title of the manuscript. Do authors have in mind that one takes wire of "mixed Weyl semimetal", connects it to electrodes and injects current into it, or something else? Figure 1(a) and bulk calculations assume infinite material with applied electric field, but in practice device in Fig. 1(a) of finite size will have longitudinal current flowing along the device edges and therefore finite resistance (e.g., $C = 1$ means resistance $R = h/2e^2$ in finite size device). For example, even voltage induced precessional switching in conventional CoFeB/MgO junctions with voltage controlled magnetic anisotropy, where thick MgO barrier really prevents flow of longitudinal current, always has nonzero switching energy ~ 10 fJ.

Reply: We thank the reviewer for this insightful remark on the dissipationless nature of the magnetization switching. We agree that the low-power electrical control of ferromagnetism in the mentioned CoFeB/MgO magnetic tunnel junctions is limited by the technological challenge of Ohmic losses in terms of small leakage currents across these pseudo-capacitor devices [see, e.g., Appl. Phys. Lett. **108**, 012403 (2016)]. In the insulators that we study in our manuscript we also anticipate a non-zero switching energy owing to the magnetic anisotropy barrier. However, when referring to "dissipationless" we mean that any Ohmic losses in the insulating materials are drastically suppressed as a consequence of minimal electrical conductivity, in sharp contrast to metallic ferromagnets. Moreover, as we emphasize in our manuscript, the large magneto-electric coupling is not limited to the Chern insulator phase but is also prominent in the accompanying trivial phase, and we expect that optimizing a corresponding switching protocol can reduce dissipation in the latter case even further by taking into account the suppressed magnetic damping in insulators. We have added an according remark to the manuscript.

Reviewer #1: "Mixed Weyl semimetal" is not a standard concept or terminology, compared to widely used Weyl semimetals of type I and type II. On the other hand, the authors devote only one inline equation and second row in Fig. 1 to it, which is difficult to follow for either "spintronics" or "topological" community reader.

Reply: We thank the reviewer for this helpful comment. We aimed at introducing the concept of mixed Weyl semimetals in close analogy to the well-known terminology of conventional Weyl semimetals. In order to emphasize this point even more, we have modified the corresponding part of the results section.

Reviewer #1: The manuscript suggest to search for possible realizations of predicted effect among quantum anomalous Hall (QAH) systems fabricated to date, and they also advertise "mixed Weyl semimetals lay out extremely promising vistas in room-temperature applications" where room temperature is indeed crucial for applied spintronics. However, all QAH systems fabricated to date operate far below room temperature due to either too small gap driven by spin-orbit coupling or, most importantly, too small Curie temperature for quasi-2D magnets.

Reply: We absolutely agree with the reviewer that quantum anomalous Hall (QAH) systems fabricated to date do not yet operate at room temperature due to the mentioned material challenges. We are convinced that these issues can be overcome in the future and eventually the QAH effect will be observed experimentally even at room temperature. Our confidence roots in several theoretical suggestions of promising material candidates [such as PRL **110**, 116802 (2013), PRB **90**, 121103(R) (2014), PRL **117**, 056804 (2016), and many more] that are perceived to display the effect at much higher temperatures. Nevertheless, while the reviewer draws a contradicting connection between us motivating the search for large magneto-electric coupling in QAH systems fabricated to date (at very low temperatures) and possible room temperature applications that we point out, we only identify in our manuscript this class of readily available materials as one promising realization of mixed Weyl semimetals, without reference to room temperature in this context. On the other hand, our proposed mechanism indeed holds great prospects for room temperature applications of general metallic or insulating spintronic devices since in pointing out the importance of the monopoles in the electronic structure of mixed Weyl semimetals we are not limited to QAH systems per se. In fact, we show that even in the trivial insulating state of a mixed Weyl semimetal, achieved for some magnetization direction, and not accompanied by any sort of currents, the magneto-electric coupling is also remarkably strong.

Reviewer #1: The topological phase transition, from topologically non-trivial to trivial phase, as a function of magnetization direction is discussed in the manuscript and is illustrated by example in Supplementary Figure 1 assuming that magnetization in the plane leads to trivial phase. However, this is not valid for arbitrary system where in-plane magnetization together with broken symmetries can lead to topologically non-trivial phase, see Phys. Rev. B **94**, 085411 (2016).

Reply: We thank the referee for this comment: we agree that certain film systems can display fascinating non-trivial phases even under an in-plane magnetization direction. The general mechanism that we propose in our manuscript relies only on the emergence of monopoles but not at all on the fact that we use as an example a system with trivial topology for an in-plane magnetization. These monopoles as key ingredients appear at

any phase boundary between two topologically distinct phases. In fact, this applies also to the interesting reference Phys. Rev. B **94**, 085411 (or also arXiv: 1706.01851) where the authors find several distinct non-trivial phases in buckled honeycomb monolayers upon rotating the magnetization direction within the film plane. Given inversion asymmetry, we anticipate no fundamental obstructions for the latter systems to exhibit strong anti-damping spin-orbit torques according to the mechanism discussed in our manuscript.

Reviewer #2: Hanke, et al in their manuscript NCOMMS-17-09884 “MixedWeyl semimetals and dissipationless magnetization control in insulators by spin-orbit torques” employed both tightbinding models and ab initio electronic structure calculations for the following two-dimensional systems: (1) Magnetically doped graphene; (2) Semi-hydrogenated Bi (111) bilayer; and (3) GaBi thin film of triangular structure under an external magnetic field. In the first two cases, the transition from the magnetic trivial to non-trivial insulator is induced by varying the direction of the magnetization while in the latter the transition from the non-magnetic trivial to a magnetic non-trivial insulator is induced by the external magnetic field. The authors show that the strength of the spin-orbit torque and the Dzyaloshinskii-Moriya interaction in the non-topologically nontrivial magnetic insulator is enhanced by about a factor of 4 to 5 compared to the ferromagnetic/heavy metal bilayers. The authors propose that the underlying mechanism lies on the emergence of mixed Weyl semimetal phase in which the magnetic monopoles appear in the composite momentum direction-momentum space topology. The magnetization switching via spin-orbit torque has attracted a lot of attention recently because it has great potential in ultralowpower magnetic random access memory devices. In particular, I find it very interesting that magnitude of the spin orbit torque in insulating phases of a mixed Weyl semimetal is driven by the presence of the mixed monopole rather than the spin-orbit strength itself.

The results are interesting and the proposed underlying mechanism is novel. This work hopefully will stimulate more experimental and theoretical research and has the potential to increase the spin-orbit torque efficiency more than 3-5 times. Consequently, the manuscript does warrant publication in Nature Communications after the authors respond to the following comments and revise the manuscript accordingly.

The authors state in the Introduction that “The Berry phase nature of the anti-damping SOT manifests in the fact that the tensor elements τ_{ij} are proportional to the mixed Berry curvature of all occupied states[13,14], which incorporates derivatives of lattice-periodic wave functions u_{kn} with respect to both crystal momentum \mathbf{k} and magnetization direction \mathbf{m} .” The authors should also refer to Lee et al, Phys. Rev. B **91**, 144401 (2015) where the mixed Berry curvature over all occupied states is derived in Eqs. (14) and (15). The interesting point addressed in the above paper is that the authors show that the Fermi sea contribution can be converted to a form where the net contribution is evaluated only at the Fermi surface [Eq. (16)]. Can the authors comment on this and revise the manuscript accordingly. Namely, one would expect that only the states around the Fermi surface contribute to the current-induced anti-damping torque and not states far away from the Fermi surface.

Reply: We thank the referee for this remark, and we have included the reference to this interesting work, which we have missed out in the initial version of our manuscript. To answer the reviewer’s comment, we would like to draw the analogy to the closely related anomalous Hall conductivity, and in particular to the work PRL **93**, 206602 (2004) by Haldane. As shown in this work, although the non-quantized part of the anomalous Hall conductivity originates from the Fermi surface, it is well known that insulators without Fermi surface can still display phenomena such as quantum spin Hall and quantum anomalous Hall effect. In complete analogy, the absence of a Fermi surface does not imply the vanishing of anti-damping spin-orbit torques (which are in the focus of our study) in insulating materials (while only field-like torques would be strongly suppressed). However, we agree with the reviewer that primarily states around the insulating band gap contribute to the anti-damping spin-orbit torque, which we have stressed even more explicitly in the revised manuscript.

Reviewer #2: For the model of magnetically doped graphene that authors state “One of these monopoles appears near the K-point off any high symmetry line if the magnetization is oriented in-plane along the x-direction”. This is also shown in Fig. 1(d) where the monopole appears at $\theta = 90^\circ$ and $(k_x, k_y) = (0.29 \frac{2\pi}{a_x}, 0.41 \frac{2\pi}{a_y})$. Thus, in Fig. 2(c), (d) and (f) where the magnetization is out of plane there are no mixed Weyl nodes. How, can the authors claim that the large torkance of $0.1ea_0$ in the insulating phase arises from the monopole-driven momentum-space distribution? Indeed the torkance shows three-fold symmetry peaks around the K point but the mixed Weyl points occur only if $\theta = 90^\circ$.

Reply: We thank the reviewer for this comment. We point out that the torkance τ_{yx} does not reveal three-fold rotational symmetry since the symmetry between k_x and k_y is naturally broken in the underlying Berry phase expression. Moreover, as we outline in the subsection “Proof of monopole-driven SOT enhancement”, we explicitly verify that the large torkance in the case of an out-of-plane magnetization is uniquely identified with the mixed Weyl point that occurs at $\theta = 90^\circ$. By removing this electronic-structure feature through the inclusion of additional terms in the model such as an intrinsic spin-orbit coupling or on-site terms breaking the sub-lattice symmetry, we find that the torkance vanishes. Thus, we demonstrate unambiguously that the mixed Weyl point at $\theta = 90^\circ$ is felt even by the Berry curvature to an out-of-plane magnetization direction.

Reviewer #2: The statement in the caption of the bottom of Fig. 2(b) states “... in the (k_x, θ) -space. What are the values of k_x and θ where the gap closes at $k_y = 0.08 \frac{2\pi}{a_y}$ in the figure? The authors should clarify in the main text the statement “ (k_x, θ) -space”.

Reply: The gap closing depicted at the bottom of Fig. 2(b) occurs for $\theta = 270^\circ$ and $(k_x, k_y) = (0.71 \frac{2\pi}{a_x}, 0.09 \frac{2\pi}{a_y})$. We have added this information to the caption of Fig. 2, and have further clarified the term “ \$(k_x, \theta)\$ -space” in the captions of Figs. 2 and 3.

Reviewer #2: In order to determine the accurate evolution of the energy gap using the ab initio calculations one needs an exchange correlation functional which yields accurate values of the band gap (for example MBJLDA). I assume that the authors employed the PBE functional and hence the emergence of the Weyl points in mixed space is not very accurate. The authors should mention the type of exchange correlation they used and a statement that the emergence of the Weyl points may not be that accurate.

Reply: We completely agree with the referee that the accurate prediction of energy band gaps within density functional theory can be influenced by the choice of the exchange correlation functional. While we used PBE in the manuscript, calculations with other functionals gave the same Weyl points but at slightly different positions. We have added an according statement to the methods section. None of our general conclusions on the monopole origin of anti-damping spin-orbit torques in mixed Weyl semimetals are affected by the choice of the functional.

Reviewer #2: It is interesting that the Dzyaloshinskii-Moriya interaction shows a linear variation with energy in all Figs. 1(d), 2(f) and 4(e) and changes sign where the torkance is constant within the specific energy window. What is the underlying origin?

Reply: As we briefly mention in the subsection on the graphene model, the origin of the linear energy dependence of the Dzyaloshinskii-Moriya interaction lies in its Berry phase expression given in the methods section. Taking the derivative of this expression with respect to the Fermi energy, we find that the result is determined by the torkance. Since the latter quantity takes a constant finite value throughout the band gap, the DMI changes linearly with energy, and can even change sign. It is interesting to note that this relation is strongly reminiscent of the well-known connection between orbital magnetization and anomalous Hall conductivity in Chern insulators, which is discussed in the cited references [13], [36], and [37].

Reviewer #2: Is the Ω_{yy}^{mk} in the caption of Fig. 3 a typo? Why this is not shown in the figure?

Reply: The mixed curvature Ω_{yy}^{mk} in the caption of Fig. 3 is not a typo but one of the three components of the curvature field that we display in Fig. 3 (c) by grey arrows in the vicinity of the monopole. In addition, we make use of colors to show the field components Ω_{xy}^{kk} and Ω_{yx}^{mk} in order to emphasize the monopole-like nature of the full curvature field. We are convinced that this design contains all necessary information to make the reader aware of the distribution of the Berry curvature field near the monopole. Therefore, we do not display the color-coded values of Ω_{yy}^{mk} in a third panel.

Reviewer #2: The statement “...torkance amounts to a huge value of $-2.9ea_0$...” needs to be tone down. A factor of five is not huge.

Reply: Following the reviewer’s suggestion to tone down the statement at question, we have changed the sentence into “... amounts to a very large value ...”. We are convinced that this choice is well suited to set our substantially larger result into the context of typical torkances of metals on the order of $0.5ea_0$.

Reviewer #3: This manuscript reports a potentially interesting concept for generating large spin-orbit torque and DMI in mixed Weyl semimetals. The SOT and DMI have been calculated within linear response frame by evaluating Berry phase for materials include W-doped Graphene and H-GaBi. It is found that dependent on the magnetic axis when the two materials are in the topology phase a large SOT and DMI will arise in both materials. The concept of generating huge SOT and DMI in insulating mixed Weyl materials seems attractive for spintronics application and the manuscript is therefore can be published in nat.comm.

However there are some concerns if the authors can address.

The authors did not discuss the magnetic anisotropy of the proposed materials for example what is the magnetic ground state of the W-doped Graphene and H-GaBi? In the cited paper (PRL **108**, 056802), it is shown the W-doped Graphene will show in-plane magnetic anisotropy which means its ground state is trivial and not the mixed Weyl phase. Then experimentally how to make the proposed materials into mixed Weyl semimetals phase? An extra out-of-plane magnetic field or gate voltage will be needed to drive the materials into topological phase? As it is shown in the paper by Zhang et. al (PRL **108**, 056802), by using an extra gate voltage may be a better choice. It is better if this point can be addressed in the revised manuscript.

Reply: We thank the reviewer for this remark. Indeed, W-doped Graphene shows an in-plane anisotropy of $|E_{\text{mae}}| = 1.1$ meV whereas the functionalized bismuth bilayer prefers an out-of-plane magnetization with $|E_{\text{mae}}| = 0.8$ meV. These values are directly comparable to those in metallic heterostructures of heavy metals on ferromagnets such as Co/Pt as we have stated in the corresponding paragraph. In order to address the raised point, we remind the reviewer that a mixed Weyl semimetal describes not a system with a single magnetization direction but refers rather to an ensemble of all possible directions θ . It is in this sense that W-doped graphene and H-Bi are mixed Weyl semimetals. While the considered mixed Weyl semimetal materials could be insulating or metallic depending on the magnetization direction, special band crossings (monopoles) are hosted for certain values of θ . In our manuscript, we demonstrate unambiguously that enhanced magneto-electric properties originate from these band crossings in mixed Weyl semimetals. As a result of these monopoles in the underlying composite phase space, a large response manifests even far away from the monopole, i.e., even for a general magnetization direction for which the system is an insulator (trivial or non-trivial) or a metal. Accordingly, to observe the proposed enhancement, it is not at all necessary to tune the magnetization direction exactly to the monopole point by using the suggested means.

Reviewer #3: Another issue is about the torque produced by magnetic anisotropy. At as we know there is also a torque produced by the magnetic anisotropy when the magnetic axis is rotating away from the easy axis. What's the relation between this magnetic anisotropy torque and the SOT discussed in the present manuscript?

Reply: We thank the reviewer for this question on the magnetic anisotropy torque, which is of great technological relevance for non-volatile information storage in general. From the viewpoint of symmetries, the anti-damping spin-orbit torque induced by the electric field and the torque originating from the magnetic anisotropy are qualitatively very distinct. As the magnetic anisotropy torque is not determined by a Berry curvature, it will be not susceptible to the presence of mixed Weyl points (monopoles) hidden in the electronic structure. Evaluating quantitatively the magnetic anisotropy torque, we find that its magnitude in all considered materials is comparable to typical values known

for metallic heterostructures [see, e.g., PRB **90**, 174423 (2014)]. Taking as an example the functionalized bismuth bilayer with $\theta = 30^\circ$, we estimate that the magnitudes of the magnetic anisotropy torque and the anti-damping torque become equal for an applied in-plane electric field with the strength of about $5 \text{ mV}/\text{\AA}$. We have added a corresponding paragraph addressing this issue to the revised manuscript. We also point out that the linear dependence on the SOT on the electric field opens an intriguing perspective to tune the anti-damping torque over wider range of values, given the insulating nature of the material under consideration.

Reviewer #3: The torque has been calculated when the electric field is applied in-plane and in principle there is also SOT when electric field is vertically applied as in Zhang's paper. Then under which electric field the SOT will be more efficient?

Reply: Within our theoretical formalism, we are not only able to determine the anti-damping spin-orbit torque as response to an in-plane electric field (as mediated, e.g., by the torkances τ_{yx} and τ_{yy}), but we may also estimate the torque as response to an out-of-plane electric field. However, we find such spin-orbit torques to be much less efficient under the same field strength, for example, the corresponding torkance τ_{yz} is typically several orders of magnitude smaller than the aforementioned τ_{yx} and τ_{yy} . We therefore conclude that the spin-orbit torques generated by in-plane electric fields are the most relevant ones for magnetization switching in the considered thin films.

Reviewer #3: A minor comment. It is better that the Cartesian coordinates are shown in Fig.1a and b, which may describe the directions of electric field, torque, DMI more clearly. And also the torque direction in Fig.1b is shown to be in the xz plane, however in the supplementary (Fig.1) the torque components are in xy plane?

Reply: We agree with the reviewer and have clarified the directions of all shown quantities in Figs. 1 (a) and (b) by adding a Cartesian coordinate system. The reviewer is also correct that the torque shown in the general overview Fig. 1(b) is chosen to reside in the xz plane perpendicular to the electric field. Possible torque components parallel to the applied field are not illustrated for clarity. On the other hand, the calculated torkances τ_{yx} and τ_{yy} that we display in Supplementary Figure 1 mediate spin-orbit torques that point along the y axis upon applying an electric field along x or y direction.

Reviewers' Comments:

Reviewer #1 (Remarks to the Author):

The author have made substantial effort to respond to all three referees questions and improve the manuscript, which can now be published. However, their insistence on idiosyncratic definition of "dissipationless" collides with standard terminology employed in experimental and engineering spintronic literature where one often finds table of energies required for magnetization switching, and proposed control is certainly not going to happen with 0 J dissipation. Also, it is not clear what is meant by the explanation in the Discussion section about "vast suppression of any Ohmic losses, the minimization of which is well known to set a central challenge for magnetotransport in topological insulators and quantum Hall systems" since realistic device made of material with, e.g., Chern number C will inevitably generate Ohmic dissipation $I^2 R$, $R = h/(2e^2 C)$ that is insensitive to increasing disorder and can only be further minimized (but never made zero) by increasing Chern number. In other words, I suggest to remove confusing "dissipationless" from the title and/or replace with more modest "low-dissipation" or "ultralow dissipation" that would be understandable to experimental and engineering communities.

Reviewer #2 (Remarks to the Author):

The authors have responded satisfactorily to all comments, revisions, and suggestions and have revised the manuscript accordingly. The manuscript warrants publication in Nature Communications.

Reviewer #3 (Remarks to the Author):

After reading the response letter and the revised manuscript, I think the authors have correctly addressed all the comments raised by the referees and make appropriate changes. Thus I recommend that it can be accepted for publication in nat.comm.

Reviewers' Comments:

Reviewer #1: The author have made substantial effort to respond to all three referees questions and improve the manuscript, which can now be published. However, their insistence on idiosyncratic definition of "dissipationless" collides with standard terminology employed in experimental and engineering spintronic literature where one often finds table of energies required for magnetization switching, and proposed control is certainly not going to happen with 0 J dissipation. Also, it is not clear what is meant by the explanation in the Discussion section about "vast suppression of any Ohmic losses, the minimization of which is well known to set a central challenge for magnetotransport in topological insulators and quantum Hall systems" since realistic device made of material with, e.g., Chern number C will inevitably generate Ohmic dissipation I^2R , $R = h/(2e^2C)$ that is insensitive to increasing disorder and can only be further minimized (but never made zero) by increasing Chern number. In other words, I suggest to remove confusing "dissipationless" from the title and/or replace with more modest "low-dissipation" or "ultralow dissipation" that would be understandable to experimental and engineering communities.

Reply: Following the Reviewer's helpful remark, we realize that the used terminology at question might be misleading for experimental and engineering communities, and we have thus replaced "dissipationless" with the suggested "low-dissipation" throughout our manuscript (including the title). As this replacement renders the unclear explanation in the Discussion section redundant, we have removed the corresponding sentence. We would like to thank the Reviewer for his thorough reading of the manuscript and for the requests for clarification, which have improved the final form of the manuscript.

Reviewer #2: The authors have responded satisfactorily to all comments, revisions, and suggestions and have revised the manuscript accordingly. The manuscript warrants publication in Nature Communications.

Reply: We would like to thank the Reviewer for all valuable suggestions, and for considering our work as suitable for publication.

Reviewer #3: After reading the response letter and the revised manuscript, I think the authors have correctly addressed all the comments raised by the referees and make appropriate changes. Thus I recommend that it can be accepted for publication in nat.comm.

Reply: We would like to thank the Reviewer for all insightful comments as well as for recommending our work for publication.